

# Novel, non-symbiotic isolates of *Neorhizobium* from a dryland agricultural soil

Amalia Soenens[1] and Juan Imperial[1,2]

[1] Centro de Biotecnología y Genómica de Plantas, Universidad Politécnica de Madrid-Instituto Nacional de Investigación y Tecnología Agraria y Alimentaria, Pozuelo de Alarcón, Madrid, Spain
[2] Instituto de Ciencias Agrarias, Consejo Superior de Investigaciones Científicas, Madrid, Spain

## ABSTRACT

Semi-selective enrichment, followed by PCR screening, resulted in the successful direct isolation of fast-growing Rhizobia from a dryland agricultural soil. Over 50% of these isolates belong to the genus *Neorhizobium*, as concluded from partial *rpoB* and near-complete 16S rDNA sequence analysis. Further genotypic and genomic analysis of five representative isolates confirmed that they form a coherent group within *Neorhizobium*, closer to *N. galegae* than to the remaining *Neorhizobium* species, but clearly differentiated from the former, and constituting at least one new genomospecies within *Neorhizobium*. All the isolates lacked *nod* and *nif* symbiotic genes but contained a *repABC* replication/maintenance region, characteristic of rhizobial plasmids, within large contigs from their draft genome sequences. These *repABC* sequences were related, but not identical, to *repABC* sequences found in symbiotic plasmids from *N. galegae*, suggesting that the non-symbiotic isolates have the potential to harbor symbiotic plasmids. This is the first report of non-symbiotic members of *Neorhizobium* from soil.

## INTRODUCTION

A group of α-proteobacteria from the Rhizobiales order, especially within the Rhizobiaceae and Bradyrhizobiaceae families, are collectively known as Rhizobia because they have the ability to establish root-nodule symbioses with legumes. Within these nodules, Rhizobia fix atmospheric nitrogen, and this fixed nitrogen is assimilated by the plant. This makes most legumes uniquely independent from the need of any exogenous nitrogen fertilizer, an important ecological and agricultural trait that is at the basis of any effort aimed at sustainable agriculture.

Since their first isolation from legume root nodules (*Beijerinck, 1888*), it has been known that Rhizobia are present in the soil, wherefrom they can colonize emerging roots of their legume host. However, they have rarely been isolated directly from the soil, given that the use of trap legume plants provides a facile method for Rhizobia isolation from root nodules. This is very convenient, especially because rhizobial soil populations have been often estimated as ranging between $10^2$–$10^5$ cells per gram of soil, depending on soil type

Corresponding author
Juan Imperial,
juan.imperial@upm.es,
juan.imperial@csic.es

and host plant (*Singleton & Tavares, 1986*). However, the use of legume trap plants allows a very limited glimpse at rhizobial populations in soil, for two reasons. First, legume-rhizobial symbioses are usually very specific, and a specific legume can only be nodulated by a specific type of Rhizobium, a phenomenon whose molecular bases have been intensively studied in recent years. Second, the genetic determinants for a successful symbiosis with a host legume are but a small fraction of the genetic complement of a Rhizobium. These determinants are often present in plasmids or mobile genomic islands that can be transferred, exchanged or lost (*López-Guerrero et al., 2012*; *Andrews & Andrews, 2017*). Therefore, it is possible that, for any given Rhizobium, a large non-symbiotic subpopulation co-exists with the symbiotic subpopulation and represents an ill-studied reservoir of genetic diversity. That this is indeed the case has been demonstrated in very few instances. In a pioneering study, *Sullivan et al. (2002)* were able to show that *Mesorhizobium loti*, the microsymbiont of *Lotus* spp. harbors its symbiotic determinants within a symbiotic genetic island integrated in its chromosome, and that this island can excise and be transferred to other cells. Furthermore, after inoculating a soil with a strain containing a marked symbiotic island, they were able to recover it in different genomic backgrounds, thus proving that the island undergoes cell-to-cell transfer in the soil, and that non-symbiotic *M. loti* strains are present in the soil that can receive the marked symbiotic island and thus acquire the ability to nodulate *Lotus*.

Despite the above, the direct isolation of Rhizobia from soil has received little attention, and even in those few cases, the interest was placed on symbiotically-competent Rhizobia. For example, *Louvrier, Laguerre & Amarger (1995)*, in ground-breaking work, devised a semi-selective culture medium to enrich *Rhizobium leguminosarum* from soil. They were interested in isolating symbiotic strains that had not been selected by the plant host, in order to test the hypothesis that the different plant host species this bacterium colonizes, select specific genotypes among those present in the soil (*Louvrier, Laguerre & Amarger, 1996*). Likewise, Tong and Sadowsky optimized a *Bradyrhizobium japonicum* and *B. elkanii*—enriching medium (1994) that was later used to isolate symbiotic and non-symbiotic Bradyrhizobia from soil (*Pongsilp et al., 2002*). In our lab, we have recently built upon Laguerre and Amarger's observations by carrying out a population genomics study of genotype selection by the legume host in the same agricultural soil in Burgundy (*Jorrín & Imperial, 2015*). In this study, many non-symbiotic Rhizobia were isolated (*Jorrín, 2016*). The above results are in line with the recent identification of non-symbiotic Rhizobia as abundant components of plant microbiomes (*Lundberg et al., 2012*; *Shakya et al., 2013*; *Chaparro, Badri & Vivanco, 2014*; *Ofek-Lalzar et al., 2014*; *De Souza et al., 2016*) and the isolation of non-symbiotic Rhizobia from rhizospheric soil (*Segovia et al., 1991*; *Sullivan et al., 1996*; *Van Insberghe et al., 2015*; *Jones et al., 2016*) and the surface of roots (*Sullivan et al., 1996*).

In this work, we set out to directly isolate Rhizobia from a dryland agricultural soil in Southern Spain where no record of legume cultivation is available. A large fraction of those isolates was found to constitute a hitherto unsuspected, non-symbiotic clade within the recently described genus *Neorhizobium* (*Mousavi et al., 2014*), whose known members were, up to this work, legume symbionts: *N. galegae*, isolated as symbionts of the cold-climate legume *Galega* sp. (*Lindström, 1989*) as well as of many other legumes;

**Table 1  Bacterial strains used in this study.**

| Strain | Relevant characteristics[a] | Reference or Source |
|---|---|---|
| *N. alkalisoli* DSM 21826[T] | Type strain, Nod$^+$Fix$^+$, LB$^-$ | *Mousavi et al. (2014)* |
| *N. galegae* HAMBI 540[T] | Type strain, Nod$^+$Fix$^+$, LB$^-$ | *Mousavi et al. (2014)* |
| *N. huautlense* DSM 21817[T] | Type strain, Nod$^+$Fix$^+$, LB$^-$ | *Mousavi et al. (2014)* |
| *Neorhizobium* sp. | | |
| T4_1 | Soil isolate, Tomejil, Nod$^-$Fix$^-$, LB$^-$ | This study |
| T4_8 | Soil isolate, Tomejil, Nod$^-$Fix$^-$, LB$^-$ | This study |
| T5_2 | Soil isolate, Tomejil, Nod$^-$Fix$^-$, LB$^-$ | This study |
| T5_26 | Soil isolate, Tomejil, Nod$^-$Fix$^-$, LB$^-$ | This study |
| T5_27 | Soil isolate, Tomejil, Nod$^-$Fix$^-$, LB$^-$ | This study |
| T6_1 | Soil isolate, Tomejil, Nod$^-$Fix$^-$, LB$^-$ | This study |
| T6_21 | Soil isolate, Tomejil, Nod$^-$Fix$^-$, LB$^-$ | This study |
| T6_23 | Soil isolate, Tomejil, Nod$^-$Fix$^-$, LB$^-$ | This study |
| T6_25 | Soil isolate, Tomejil, Nod$^-$Fix$^-$, LB$^-$ | This study |
| T7_1 | Soil isolate, Tomejil, Nod$^-$Fix$^-$, LB$^-$ | This study |
| T7_7 | Soil isolate, Tomejil, Nod$^-$Fix$^-$, LB$^-$ | This study |
| T7_8 | Soil isolate, Tomejil, Nod$^-$Fix$^-$, LB$^-$ | This study |
| T7_9 | Soil isolate, Tomejil, Nod$^-$Fix$^-$, LB$^-$ | This study |
| T7_11 | Soil isolate, Tomejil, Nod$^-$Fix$^-$, LB$^-$ | This study |
| T7_12 | Soil isolate, Tomejil, Nod$^-$Fix$^-$, LB$^-$ | This study |
| T17_20 | Soil isolate, Tomejil, Nod$^-$Fix$^-$, LB$^-$ | This study |
| T8_5 | Soil isolate, Tomejil, Nod$^-$Fix$^-$, LB$^-$ | This study |
| T9_24 | Soil isolate, Tomejil, Nod$^-$Fix$^-$, LB$^-$ | This study |
| T11_12 | Soil isolate, Tomejil, Nod$^-$Fix$^-$, LB$^-$ | This study |
| T13_2 | Soil isolate, Tomejil, Nod$^-$Fix$^-$, LB$^-$ | This study |
| T16_1 | Soil isolate, Tomejil, Nod$^-$Fix$^-$, LB$^-$ | This study |
| T16_2 | Soil isolate, Tomejil, Nod$^-$Fix$^-$, LB$^-$ | This study |
| T16_4 | Soil isolate, Tomejil, Nod$^-$Fix$^-$, LB$^-$ | This study |
| T16_9 | Soil isolate, Tomejil, Nod$^-$Fix$^-$, LB$^-$ | This study |
| T16_12 | Soil isolate, Tomejil, Nod$^-$Fix$^-$, LB$^-$ | This study |
| T17_4 | Soil isolate, Tomejil, Nod$^-$Fix$^-$, LB$^-$ | This study |
| T17_6 | Soil isolate, Tomejil, Nod$^-$Fix$^-$, LB$^-$ | This study |
| T17_14 | Soil isolate, Tomejil, Nod$^-$Fix$^-$, LB$^-$ | This study |
| T17_15 | Soil isolate, Tomejil, Nod$^-$Fix$^-$, LB$^-$ | This study |
| T17_26 | Soil isolate, Tomejil, Nod$^-$Fix$^-$, LB$^-$ | This study |
| T18_15 | Soil isolate, Tomejil, Nod$^-$Fix$^-$, LB$^-$ | This study |
| T20_10 | Soil isolate, Tomejil, Nod$^-$Fix$^-$, LB$^-$ | This study |
| T20_15 | Soil isolate, Tomejil, Nod$^-$Fix$^-$, LB$^-$ | This study |
| T20_22 | Soil isolate, Tomejil, Nod$^-$Fix$^-$, LB$^-$ | This study |
| T20_25 | Soil isolate, Tomejil, Nod$^-$Fix$^-$, LB$^-$ | This study |

**Table 1** (*continued*)

| Strain | Relevant characteristics[a] | Reference or Source |
|---|---|---|
| T21_1 | Soil isolate, Tomejil, Nod⁻Fix⁻, LB⁻ | This study |
| T21_15 | Soil isolate, Tomejil, Nod⁻Fix⁻, LB⁻ | This study |
| T21_19 | Soil isolate, Tomejil, Nod⁻Fix⁻, LB⁻ | This study |
| T22_7 | Soil isolate, Tomejil, Nod⁻Fix⁻, LB⁻ | This study |
| T22_11 | Soil isolate, Tomejil, Nod⁻Fix⁻, LB⁻ | This study |
| T22_47 | Soil isolate, Tomejil, Nod⁻Fix⁻, LB⁻ | This study |
| T23_12 | Soil isolate, Tomejil, Nod⁻Fix⁻, LB⁻ | This study |
| T23_26 | Soil isolate, Tomejil, Nod⁻Fix⁻, LB⁻ | This study |
| T24_19 | Soil isolate, Tomejil, Nod⁻Fix⁻, LB⁻ | This study |
| T24_25 | Soil isolate, Tomejil, Nod⁻Fix⁻, LB⁻ | This study |
| T25_4 | Soil isolate, Tomejil, Nod⁻Fix⁻, LB⁻ | This study |
| T25_5 | Soil isolate, Tomejil, Nod⁻Fix⁻, LB⁻ | This study |
| T25_7 | Soil isolate, Tomejil, Nod⁻Fix⁻, LB⁻ | This study |
| T25_13 | Soil isolate, Tomejil, Nod⁻Fix⁻, LB⁻ | This study |
| T25_19 | Soil isolate, Tomejil, Nod⁻Fix⁻, LB⁻ | This study |
| T25_20 | Soil isolate, Tomejil, Nod⁻Fix⁻, LB⁻ | This study |
| T25_27 | Soil isolate, Tomejil, Nod⁻Fix⁻, LB⁻ | This study |
| T25_28 | Soil isolate, Tomejil, Nod⁻Fix⁻, LB⁻ | This study |
| T25_30 | Soil isolate, Tomejil, Nod⁻Fix⁻, LB⁻ | This study |
| T28_6 | Soil isolate, Tomejil, Nod⁻Fix⁻, LB⁻ | This study |

**Notes.**

[a]Nod, nodulation phenotype; Fix, nitrogen fixation phenotype; LB, growth on LB medium.

*N. alkalisoli*, from nodules of *Caragana intermedia* in Norther China (*Li Lu et al., 2009*); and *N. huautlense* from nodules of *Sesbania herbacea* (*Wang et al., 1998*).

# MATERIALS & METHODS

## Bacterial strains and growth conditions

Bacterial strains used in this work are listed in Table 1. Rhizobial strains were grown in Yeast Mannitol Broth (YMB; *Vincent, 1970*) at 28 °C, either in liquid culture or on solid media supplemented with 1.5% agar. For long-term maintenance, strains were grown at 28 °C in YMB and preserved in 20% glycerol at −80 °C.

## Soil

Tomejil soil is a dryland agricultural soil from the Las Torres-Tomejil Experimental Agricultural Station of the Instituto de Investigación y Formación Agraria y Pesquera de Andalucía (IFAPA) in Seville, Spain, where winter wheat is usually grown, cardoon occasionally, and there is no previous record of legume cultivation. A 10 m × 10 m plot (37°24′33.10″N, 5°34′51.91″W; 77 m above sea level) was selected and fenced for our research. For soil collection, portions of soil down to 30–50 cm depth were collected with a shovel from several points within the plot. Soil samples were maintained at 4°C and −20°C. Physicochemical characterization was done externally in Laboratorio de Edafología y Técnicas Analíticas Instrumentales, EUIT Agrícola, UPM, Madrid. Physicochemical properties of the soil are described in Table S1.
## Direct isolation of Rhizobia from soil

For *Neorhizobium* isolation we have used a modification of the Louvrier et al. protocol (*Louvrier, Laguerre & Amarger, 1995*) for enrichment of fast-growing Rhizobia (*Jorrín, 2016*; Fig. 1), as follows. In an erlenmeyer flask a $10^{-1}$ soil dilution was made using a salt buffer (0.1 g NaCl, 0.5 g K$_2$HPO4, 0.2 g MgSO$_4$·7H$_2$O, pH 6.8). This was shaken overnight at 28 °C at 200 rpm. Serial dilutions up to $10^{-6}$ depending on the soil, were made in the same salt buffer and 100 µl portions of each dilution were plated in the semi-selective medium MNBP (per liter: 1 g mannitol, 178.5 mg Na$_2$PO$_4$, 100 mg MgSO$_4$·7 H$_2$O, 33.25 mg FeCl$_3$·6 H$_2$O, 53 mg CaCl$_2$·2 H$_2$O, 500 mg NH$_4$NO$_3$, 100 mg ciclohexymide, 25 mg bacitracine, 3 mg penicillin G, 3.5 mg pentachloronitrobenzene, 0.5 mg biotin, 0.5 mg thiamine, 0.5 mg Ca pantothenate, 5 mg benomyl, pH 6.8; *Louvrier, Laguerre & Amarger, 1995*) supplemented with 25 ppm Congo red. Plates were left for four days at 28 °C. After the incubation period, white-pink colonies were picked with a sterile toothpick into Yeast Mannitol Broth (YMB), Luria-broth (LB; *Bertani, 1951*), and MNBP agar plates and incubated at 28 °C for two days. Negative LB colonies were then streaked out a second time into YMB and LB agar plates. Confirmed negative LB colonies were grown on YMB agar plates until pure cultures were obtained. Genomic DNA was extracted from pure cultures using the alkaline lysis method (*Baele et al., 2000*) and tested for *fnrN* by PCR amplification. Positive *fnrN* isolates were characterized phylogenetically by PCR amplification and sequencing of the housekeeping genes 16S rDNA and *rpoB*. Presence of symbiotic genes was determined by amplification and sequencing (if present) of *nodC* and *nifH*.

## Genotypic characterization

Bacterial DNA previously isolated by alkaline lysis (*Baele et al., 2000*) was used as substrate for PCR amplifications. Full-length 16S rDNA (*Weisburg et al., 1991*), partial *nodC* (*Sarita et al., 2005*) and partial *nifH* (*Ando et al., 2005*) sequences were amplified with primers described in the references. Partial *rpoB* was amplified with: F_rpoB (5′-GARTTCGACGCCAAGGAYAT-3′) and R_rpoB (5′- GAAGAACAGCGAGTTGAACAT-3′). Amplifications were carried out in 25 µl solution containing DNA (5–10 ng), 2.5 µl 10× PCR buffer containing magnesium chloride (Roche Applied Science, Penzberg, Germany), 10 µM of each dNTP, 10 µM of each primer, 1 µl DMSO and 1 U of Taq DNA polymerase (Roche Applied Science). Unincorporated primers and dNTPs were removed from PCR products with the NucleoSpin®Extract II Kit (Macherey-Nagel, Düren, Germany) or, when needed, by gel electrophoresis followed by band purification with the same kit. Sanger sequencing was carried out externally (STAB Vida, Lisbon, Portugal).

## Genomic characterization

For bacterial genome sequencing of Tomejil isolates: T20_22, T7_12, T25_27, T25_13, T6_25, and type *Neorhizobium* strains not available in public databases, bacteria were grown in Tryptone Yeast (TY; *Beringer, 1974*). The bacterial pellet obtained after centrifugation was used to extract total DNA using the CTAB method (*Feil, Feil & Copeland, 2012*). DNA quantity and quality were assessed by spectrophotometry (Nanodrop, NanoDrop Technologies, Wilmington, DE, USA) and fluorescence (Qubit, Invitrogen by Life
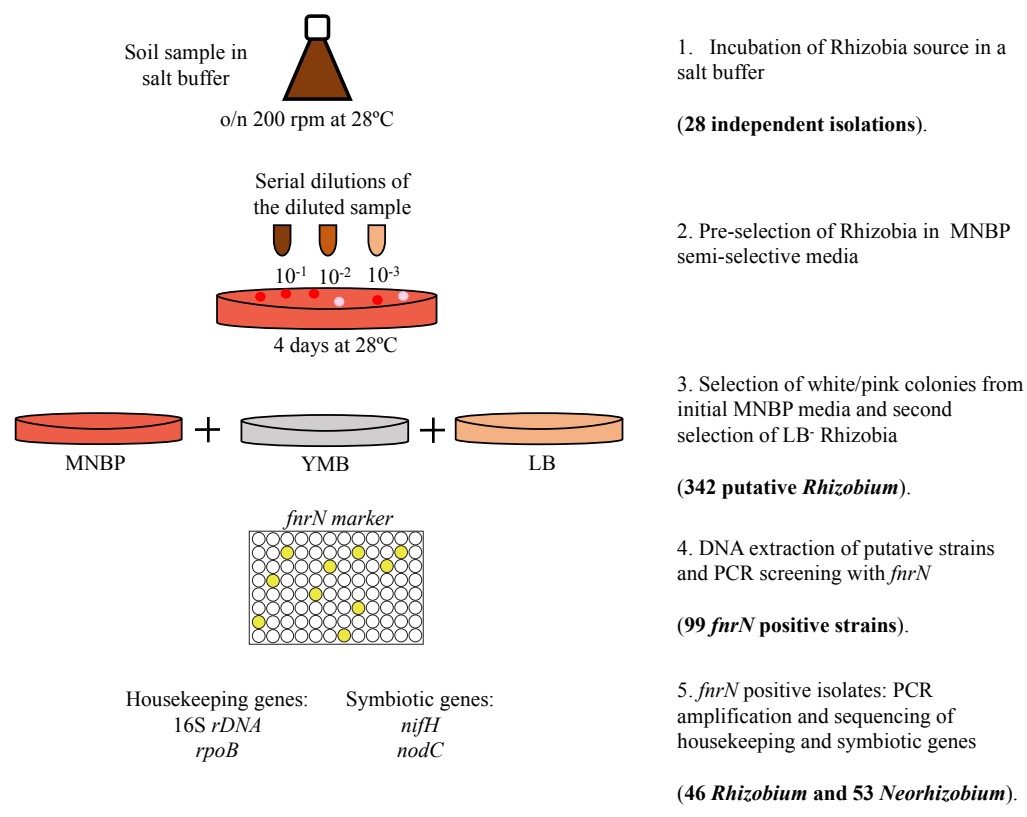

**Figure 1** Schematic representation of the semi-selective enrichment procedure to isolate fast-growing Rhizobia from Tomejil soil and the summary of results obtained.

Technologies, Singapore), and integrity and purity were checked by electrophoresis in a 0.8% agarose gel. Draft genomic sequences of bacterial strains were obtained externally (MicrobesNG, Birmingham, UK) with Illumina technology (MiSeq v3, PE 2 × 300 bp), and reads were assembled using SPAdes (*Bankevich et al., 2012*) and annotated with Prokka (*Seemann, 2014*).

## Bioinformatics

For phylogenetic and sequence analyses, nucleotide sequences obtained from PCR products were corrected and assembled if necessary with SerialCloner2-6 (http://serialbasics.free.fr/Serial_Cloner.html) and 4peaks (http://nucleobytes.com/4peaks/). Sequences were aligned with the ClustalW algorithm (*Chenna et al., 2003*) in MEGA 6.0 (*Tamura et al., 2013*). Randomized Axelerated Maximum Likelihood (RAxML) (*Stamatakis, 2014*) or MEGA 6.0 (*Tamura et al., 2013*) were used for the construction of phylogenetic trees. Phylogenetic trees were visualized with FigTree v.1.4.3 (http://tree.bio.ed.ac.uk/software/figtree/) and edited with Adobe Illustrator CS5 (Adobe Systems, San José, CA, USA). In order to study genomic identity among strains, Average Nucleotide Identity (either based on MUMmer alignments, ANIm, or based on BLAST alignments, ANIb) was calculated with the JSpeciesWS online server (*Richter et al., 2016*). A distance dendrogram was generated

by hierarchical cluster analysis of $100-\%$ ANI matrices (*Chan et al., 2012*) with StataSE v.14.0 (StataCorp, College Station, TX, USA) after computation of Euclidean distances with the Average Linked method.

### DNA sequences

GenBank accession and Bioproject numbers for sequences and genomes, respectively, obtained in this work are listed in Table S2, together with accession numbers for reference sequences used in the analyses.

## RESULTS

### Direct isolation of *Neorhizobium* sp. from soil

A total of twenty-eight independent isolations were carried out from Tomejil soil (Fig. 1). After four days, pink-white colonies appearing on MNBP plates were chosen as putative fast-growing rhizobia. From those, we chose to discard all that were able to grow on LB agar. In our hands, this includes most *Ensifer* spp. and many non-rhizobial isolates. However, most *Rhizobium* spp. were unable to grow on the 5 g l$^{-1}$ NaCl present in LB agar, and thus this step resulted in a further enrichment of *Rhizobium* spp. DNA from three hundred and forty-two of these putative *Rhizobium* spp. was amplified for the *fnrN* gene by PCR. This gene is important for microaerobic metabolism (*Gutiérrez et al., 1997*) and has been found in all sequenced *Rhizobium* spp. and *Agrobacterium* spp., but not in other Rhizobia. Ninety-nine of the DNAs resulted in good amplification of a single major band. However, two different band sizes were observed. While forty-six strains resulted in amplification of a PCR band of the expected size (279 bp), the remaining fifty-three strains amplified a larger fragment (353 bp, Fig. S1). In order to reduce the number of isolates for further characterization, a fragment of the *rpoB* gene was amplified and sequenced from all of them. The *rpoB* marker has been shown to be very effective in discriminating phylogenetically close bacteria (*Mollet, Drancourt & Raoult, 1997*; *Khamis et al., 2003*; *Case et al., 2006*), including Rhizobia (*Jorrín, 2016*). Surprisingly, *rpoB* sequences from the same fifty-three strains clustered with sequences from genus *Neorhizobium* (Fig. 2B). These sequences were compared and classified into groups (twenty in total) if they differed in at least one nucleotide (Fig. S2). In order to facilitate further studies, a representative strain of each these *rpoB* sequence types was chosen. Near complete 16S rDNA sequences were obtained from these representative strains and compared against those in databanks, confirming that they formed a diverse clade that was closely related to known members of the genus *Neorhizobium*, but distant from the *Agrobacterium-Rhizobium* group (Fig. 2A).

### Genotypic and genomic characterization of *Neorhizobium* sp. isolates

Since all previously characterized *Neorhizobium* isolates had been obtained from nitrogen-fixing legume root nodules, we tried to amplify *nodC* and *nifH*, markers related to symbiosis and nitrogen fixation, respectively, from our Tomejil isolates. Negative results were obtained in all cases, suggesting that these isolates were non-symbiotic. Since direct tests of symbiotic ability with legume plants were hampered by the fact that the known hosts of *Neorhizobium* spp. (*Galega officinalis* and *G. orientalis*, *Mousavi et al., 2014*; *Sesbania herbacea*, *Wang et*

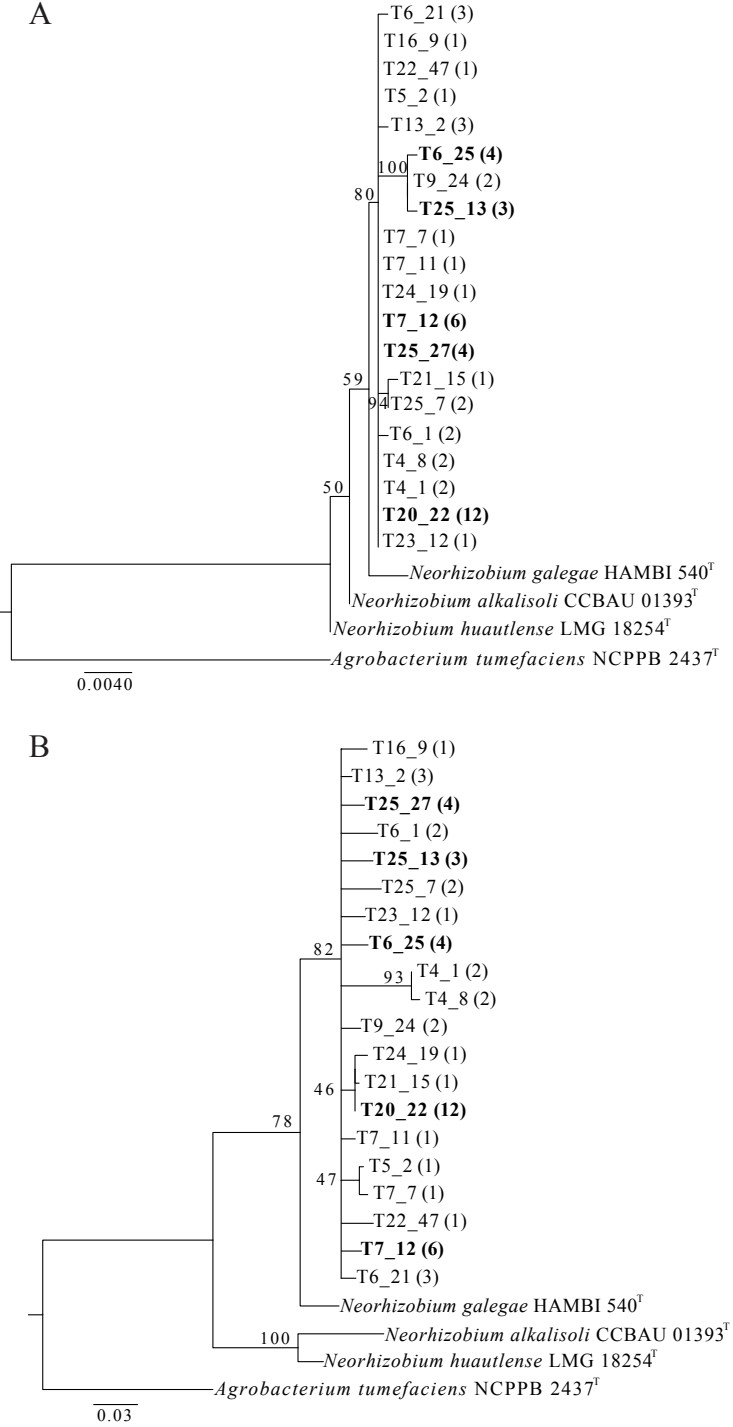

**Figure 2** **Phylogenetic tree of representative Tomejil soil isolates based on PCR amplified, near-complete 16S rDNA (1,234 bp, A) and partial *rpoB* (356 bp, B) sequences.** Maximum likelihood trees (RAxML) were derived from ClustalW alignments. T: Tomejil soil isolates representative of the different *rpoB* 

**Figure 2 (…continued)**
genotypes, as follows: T4_1 (with T18_15); T4_8 (with T16_4); T5_2; T25_27 (with T5_2, T5_26, T22_7); T6_1 (with T17_6); T6_21 (with T17_15, T22_11); T6_25 (with T8_5, T6_23, T25_30); T7_12 (with T20_25, T21_1, T21_19, T23_26, T25_20); T7_7; T7_11; T9_24 (with T20_10); T20_22 (with T5_27, T11_12, T16_2, T16_12, T17_4, T17_14, T17_20, T17_26, T24_25, T25_19, T28_6); T13_2 (with T7_1, T7_9); T16_9; T21_15; T22_47; T23_12; T24_19; T25_7 (with T16_1); T25_13 (with T25_4; T25_5). The number of strains within each genotype group is indicated within parentheses. Trees include sequences from type strains of *Neorhizobium* species as the closest taxonomic relatives, and of *A. tumefaciens*, as outgroup. Bootstrap support (1,000 replications) for the different nodes is indicated. Bars represent the number of substitutions per base. Genbank accession numbers are listed on Table S2.

**Table 2** Genomic features of *Neorhizobium* genomes sequenced in this work.

| Strain | Number of contigs | Largest contig (bp) | Total genome length (bp) | G + C (%) | N50 |
|--------|-------------------|---------------------|--------------------------|-----------|------|
| T20_22 | 37 | 1,052,711 | 6,608,977 | 61.47 | 508,270 |
| T7_12 | 52 | 1,197,185 | 6,627,103 | 61.44 | 347,929 |
| T25_27 | 27 | 1,446,028 | 6,462,352 | 61.49 | 734,252 |
| T25_13 | 42 | 721,675 | 6,322,993 | 61.56 | 419,328 |
| T6_25 | 69 | 488,027 | 6,750,064 | 61.35 | 186,129 |

*al., 1998*; *Caragana intermedia*, *Li Lu et al., 2009*) are very diverse and are not found in the Tomejil area, we reasoned that obtaining and characterizing the genome sequence of some of the Tomejil isolates, even at draft level, would be worthwhile.

We chose five representatives of the most abundant/diverse *rpoB* groups, so as to try to obtain a picture of the genomic diversity within this *Neorhizobium* sp. clade. Genomic DNA was sequenced and assembled to draft level and this assembly was used for subsequent analysis. All five isolates had similar genome size and G+C (%) composition (Table 2).

We first searched for *fnrN* genes. This was important because no *fnrN* gene has been described in *Neorhizobium* spp. isolates, and it was absent from the *Neorhizobium* spp. genomic sequences available in databanks. Using the well-characterized *R. leguminosarum fnrN* gene for BLAST comparisons, we were unable to find any relevant hit against genomic sequences from Tomejil isolates, despite the fact that the isolates were chosen because they showed clear amplification of a single (although of different size, see above) band with *fnrN* primers (Fig. S1A). Amplified PCR bands from some of the isolates were sequenced and compared with databanks. Part of the amplified region showed similarity with genes encoding a poly (3-hydroxybutyrate) depolymerase from Rhizobia, including *Neorhizobium galegae* (but not *N. alkalisoli* or *N. huautlense*). This gene, together with an upstream ORF, was present in genome assemblies of the Tomejil strains and contained sequences partially complementary to those of the *fnrN* primers that may explain the successful amplifications observed (Fig. S1B).

We then tried to find *nod* and *nif* genes by running BLAST searches of known *Neorhizobium nod* and *nif* genes against draft genome sequences, with negative results in all cases. Since *nod* and *nif* genes are harbored on large megaplasmids in *N. galegae* (Österman et al., 2014), we searched for similar plasmids in the genomes of Tomejil strains. All five genomes contained a set of *repABC* sequences characteristic of rhizobial plasmids

**Table 3   Presence of *repABC* regions in the genomes of Tomejil strains.** DNA regions similar to the 3,628 bp region containing *repABC* genes from the symbiotic megaplasmid from *N. galegae* bv. orientalis HAMBI 540 were located in genome sequences by BLAST, extracted and compared by multiple alignment (ClustalW).

| Genome | Number of *repABC* regions | (%) identity to HAMBI 540 | Size of contig (bp) |
|---|---|---|---|
| *N. galegae* bv. orientalis HAMBI 540 | 1 | 100 | 1,807,065 |
| *N. galegae* bv. officinalis HAMBI 1141 | 2 | 95 | 1,638,739 |
| | | 53 | 175,279 |
| T20_22 | 1 | 85 | 376,046 |
| T7_12 | 1 | 85 | 523,062 |
| T25_27 | 1 | 85 | 263,545 |
| T25_13 | 1 | 86 | 721,675 |
| T6_25 | 1 | 85 | 82,656 |

(*Cevallos, Cervantes-Rivera & Gutiérrez-Ríos, 2008*; *Pinto, Pappas & Winans, 2012*), highly similar (85–86%) to the *repABC* cluster from the 1.8 Mb megaplasmid from *N. galegae* bv. orientalis (Table 3) and that are probably responsible for the replication a large plasmid in each of these strains. This is supported by the fact that *repABC* homologues are present in large contigs (83–523 kb) in the Tomejil strains, despite the draft level quality of their genome sequences (27–69 contigs, Table 2). Multiple sequence alignment of these *repABC* sequences showed that plasmids from the Tomejil strains were highly related, and separated from the symbiotic plasmids from *N. galegae* bv. orientalis and *N. galegae* bv. officinalis (Fig. 3). The completely sequenced strain *N. galegae* bv. officinalis HAMBI 1140 harbors an additional megaplasmid (175 kb) that appeared to be highly unrelated to both *N. galegae* symbiotic plasmids and plasmids from Tomejil strains (Table 3, Fig. 3). Finally, it was interesting to note that, among the Tomejil genomes, only strain T25_13 harbored a complete set of conjugal genes (*tra* and *trb* genes). In the remaining strains, these genes were absent altogether (T6_25, T20_22, T25_27) or incomplete (T7_12, where sequences similar to just *trbG*, *trbI*, and *trbL* were present) (Table S3).

In order to ascertain the phylogenetic relationships among the sequenced Tomejil strains, and between this clade and the type strains of the three *Neorhizobium* species: *N. galegae*, *N. alkalisoli*, *N. huautlense*, we pulled out the complete sequences of the genes *atpD*, *glnII*, *recA*, *rpoB* and *thrC* from the genome assemblies. These genes had been used successfully to define the *Neorhizobium* genus and its species (*Mousavi et al., 2014*). Figure 4 shows a maximum-likelihood phylogenetic tree derived from a multiple alignment of the concatenated genes, rooted by using the *Agrobacterium tumefaciens* Ach5 sequences. There was very strong support for two clades within the *Neorhizobium* genus; one formed by *N. alkalisoli* and *N. huautlense*, and the other by *N. galegae* and the Tomejil strains. Four of the Tomejil strains (T7_12, T20_22, T25_13, T25_27) formed a well-supported group clearly separated from the fifth strain, T6_25, and all five, in turn, clearly separated from *N. galegae*.
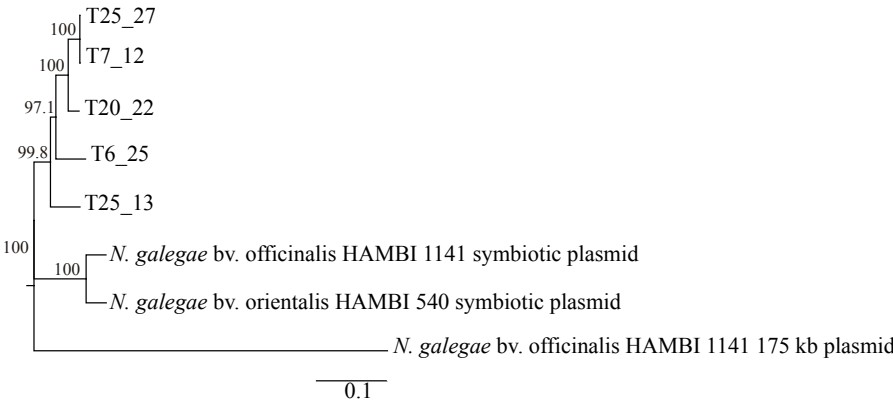

**Figure 3** **Phylogenetic tree based on *repABC* sequences from Tomejil genome sequences and from *Neorhizobium* megaplasmids.** Sequences similar to the *N. galegae* bv. orientalis HAMBI 540 1.8 Mb megaplasmid *repABC* region (3,628 bp) were extracted from Tomejil draft genomes sequences and from the *N. galegae* bv. officinalis HAMBI 1141 genome sequence, aligned with ClustalW, and a Neighbor-Joining consensus tree derived. Bootstrap support (1,000 replications) for the different nodes is indicated. Bar represents the number of substitutions per base.

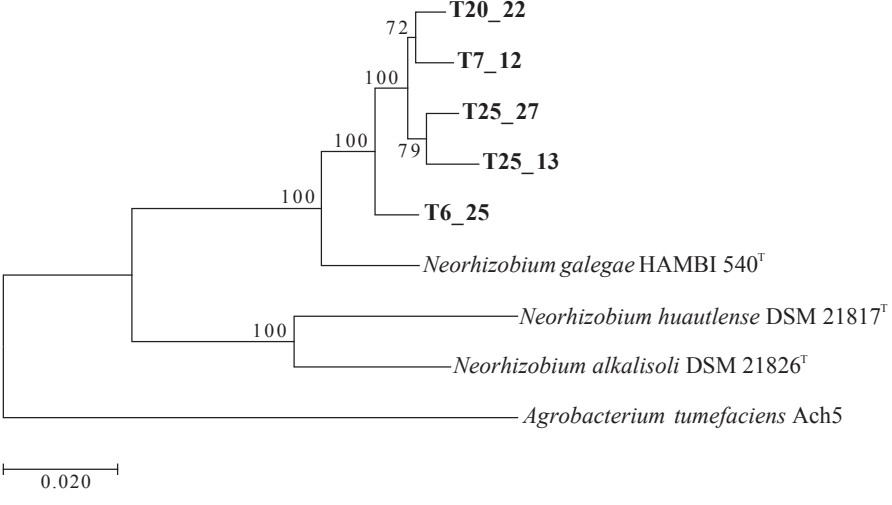

**Figure 4** **Phylogenetic tree of sequenced Tomejil strains and of *Neorhizobium* type strains based on a concatenation of complete *atpD*, *glnII*, *recA*, *rpoB*, and *thrC* genes (8,949 bp).** Maximum likelihood trees (RAxML) were derived from ClustalW alignments. The tree includes the *A. tumefaciens* type strain as outgroup. Bootstrap support (1,000 replications) for the different nodes is indicated. Bar represents the number of substitutions per base.

Genome-wide comparisons were carried out by calculating pairwise Average Nucleotide Identities (*Richter & Rosselló-Mora, 2009*; *Richter et al., 2016*) with the above genomes. Both ANIm and ANIb scores were calculated, and the ANIb matrix is shown in Table S4. A distance matrix was generated from the ANIb matrix using 100-ANIb (*Chan et al., 2012*) and Euclidean distances were calculated. These were represented in the dendrogram

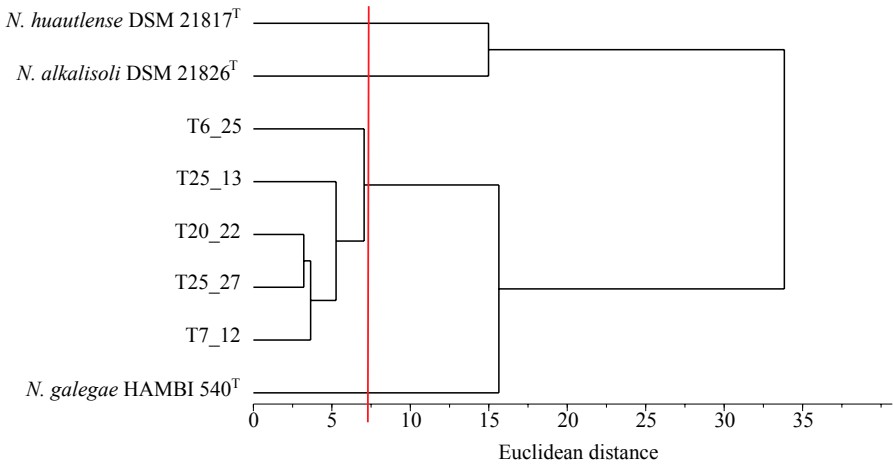

**Figure 5** Dendrogram representation of a Euclidean distance matrix derived from pairwise ANIb distances among Tomejil and *Neorhizobium* type strain genomes. The vertical red line indicates the 95% ANI threshold.

shown in Fig. 5. The dendrogram faithfully reproduced the topology of the multilocus phylogenetic tree (Fig. 4). Using a 95% ANI value as a widely accepted delimiter for genomic species (*Richter & Rosselló-Mora, 2009*), our results are consistent with Tomejil strains representing at least a clearly differentiated genospecies within *Neorhizobium*, and probably two, represented by strains T6_25 on one hand, and T7_12, T20_22, T25_13, and T25_27, on the other.

## DISCUSSION

Direct isolation of Rhizobia from soil without resorting to trapping them inside their legume host is complicated by the fact that Rhizobial populations in soil can be small (*Singleton & Tavares, 1986*) and because of the paucity of selective characters that can be used. This is especially true for non-symbiotic variants lacking any symbiotic marker. As a result, very few studies have aimed at isolating non-symbiotic Rhizobia from soil. Groundbreaking work by Sullivan and Ronson with *Mesorhizobium loti* (*Sullivan et al., 1995*) did not only reveal the existence of an abundant population of non-symbiotic variants in the soil, but also showed that the genetic determinants for symbiosis could be readily transferred to these variants within the soil. Some of these non-symbiotic variants were further characterized and suggested to be representatives of species hitherto undescribed. Using a semi-selective culture medium, *Tong & Sadowsky (1994)* were able to enrich soybean-specific Rhizobia of the species *Bradyrhizobium japonicum* and *B. elkanii*. Using this medium, the same group later described that about half of the Bradyrhizobia isolated directly from Thai soils were non-symbiotic, although they did not characterize them further (*Pongsilp et al., 2002*). *Louvrier, Laguerre & Amarger (1995)* were also able to enrich members of the genus *Rhizobium* using a specific medium. Although they focused

on symbiotic isolates (*Louvrier, Laguerre & Amarger, 1996*) they also isolated a fraction of non-symbiotic Rhizobia (G Laguerre, pers. comm., 2011).

Our group had previously used the *Rhizobium* semi-selective medium, together with the *fnrN* gene marker, to isolate Rhizobia from soil (*Jorrín, 2016*) in order to study, at the genomic level, the legume host selection of specific *R. leguminosarum* genotypes (*Jorrín & Imperial, 2015*). This phenomenon had been described by the Amarger-Laguerre (*Laguerre et al., 2003*) and the Young (*Young & Wexler, 1988*) groups. During this study, we were able to isolate a number of non-symbiotic members of the genus *Rhizobium* (*Jorrín, 2016*; *Jorrín & Imperial, 2015*). Therefore, it was surprising that the same enrichment methodology resulted in the isolation of a large (over 50%) proportion of non-symbiotic members of the genus *Neorhizobium* from soil samples from the IFAPA Tomejil Experimental Station in Carmona (Seville, Spain), first because non-symbiotic *Neorhizobium* had not been previously isolated, and second because our genetic screening with *fnrN* should have left them behind (although the different size of the PCR amplified bands was already an indication that these were atypical members of the genus *Rhizobium* at best). The serendipitous reason why these particular *Neorhizobium* were chosen as putative *Rhizobium* has been presented above. Clearly, they grow well on MNBP semi-selective medium, with cultural characteristics similar to those of *Rhizobium*. However, it is possible that, among the two hundred and forty-three colonies that tested negative for *fnrN* amplification, other *Neorhizobium* that do not have a high enough conservation of the poly (3-hydroxybutyrate) depolymerase region exist in the Tomejil soil.

At any rate, *Neorhizobium* appears to be at least as abundant in the particular sampled soil as *Rhizobium* and, given the high rates of horizontal gene transfer that can take place in soil (*Sullivan et al., 2002*), they may become relevant in the establishment of symbioses with native legumes after receiving the appropriate symbiotic genes. Acquisition of these genes might be facilitated by the fact that *N. galegae* has been shown to harbor symbiotic genes on a megaplasmid (*Radeva et al., 2001*; *Österman et al., 2014*). In view of their size, nucleotide composition and conservation, it is possible that these plasmids are chromids, as defined by *Harrison et al. (2010)*, although it has yet to be shown that they carry core genes. As shown above, all the sequenced Tomejil strains contain a set of *repABC* genes characteristic of plasmids from the Rhizobiales (*Cevallos, Cervantes-Rivera & Gutiérrez-Ríos, 2008*; *Pinto, Pappas & Winans, 2012*) in large contigs, suggesting that these strains also harbor a megaplasmid (or chromid) and are, thus, probably able to receive similar, large plasmids containing symbiotic genes.

A final consideration is whether the existence of populations of non-symbiotic *Neorhizobium* in soils reflects a normal situation in agricultural soils. This would require a large screening that is beyond the scope of this work. However, our preliminary studies suggest that in at least two other soils from Southern Spain that we have tested, members of *Neorhizobium* are not present in detectable numbers, and that the predominant isolates resulting from our semi-selective screening are *Rhizobium* spp. (A Soenens & J Imperial, 2017, unpublished data). This would then shift the question to why the Tomejil soil harbors a *Neorhizobium* population, a question that would require a better understanding of the ecology of this group in soil.

## CONCLUSIONS

In conclusion, our work has allowed, for the first time, the isolation and identification of non-symbiotic members of the genus *Neorhizobium* from soil. Genotypic and genomic characterization of these isolates suggests that they are representatives of one, or perhaps two, new genospecies within *Neorhizobium.* It also suggests that soils harbor a large diversity of Rhizobia in the form of non-symbiotic variants that have traditionally escaped characterization and that may play an important role in the biology of these organisms.

## ACKNOWLEDGEMENTS

We thank Francisco Temprano, Dulce Rodríguez-Navarro, and Francisco Perea, from IFAPA "Las Torres-Tomejil," for facilitating our soil sampling in the Tomejil experimental farm.

### Funding

This work was supported by the CONSOLIDER-INGENIO program (CSD2009-00006) and by project LUPIGEN (CGL2011-26932), both from the Spanish Plan Nacional de Investigación Científica y Técnica. The funders had no role in study design, data collection and analysis, decision to publish, or preparation of the manuscript.

### Grant Disclosures

The following grant information was disclosed by the authors:
CONSOLIDER-INGENIO program: CSD2009-00006.
LUPIGEN: CGL2011-26932.
Spanish Plan Nacional de Investigación Científica y Técnica.

### Competing Interests

The authors declare there are no competing interests.

### Author Contributions

- Amalia Soenens conceived and designed the experiments, performed the experiments, analyzed the data, contributed reagents/materials/analysis tools, prepared figures and/or tables, authored or reviewed drafts of the paper, approved the final draft.
- Juan Imperial conceived and designed the experiments, analyzed the data, contributed reagents/materials/analysis tools, prepared figures and/or tables, authored or reviewed drafts of the paper, approved the final draft.

### Data Availability

The 16S rDNA and *rpoB* sequences described here were deposited in GenBank under accession numbers MG957216 to MG957235, and MG966473 to MG966492, respectively. Genome sequences were deposited in GenBank and are listed in Table S2.

## Supplemental Information

Supplemental information for this article can be found online at http://dx.doi.org/10.7717/peerj.4776#supplemental-information.

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
