# Peer review of "Novel, non-symbiotic isolates of Neorhizobium from a dryland agricultural soil"

_PeerJ, doi:10.7717/peerj.4776_

## Round 0.1 · original submission · Minor Revisions

Dear Dr. Imperial,

I also read your manuscript with interest. I like it but I agree with the issues raised by the two expert reviewers.

I am looking forward to receiving your revised version soon.

Best regards,

Elisabeth Grohmann
Academic Editor

·

Basic reporting

The references by Segovia et al., 1991; Sullivan et al., 1996; Van Insberghe et al., 2015; Jones et al., 2016, are cited referring to "isolation of non-symbiotic Rhizobia from plant material". However, these papers are about the isolation of non-symbiotic rhizobia from rhizospheric soil. Only in the work by Sullivan bacteria were also isolated from the surface of roots. I think this should be clarified.

Experimental design

No comment

Validity of the findings

My only concern is that if megaplasmids are present in the described strains, the genomic analysis could be biased by not separating chromosomal from plasmid sequences.

Additional comments

In this paper the authors report the finding of non-symbiotic Neorhizobium strains in a soil of a region in Spain. This finding was serendipitous, because the selection procedure was directed to the identification of rhizobium species. The authors analyzed different aspects of the strains, including a partial sequence of some representatives. This analysis showed that the strains belong to a specific group, close to but different from N galegae. The possible presence of plasmids is suggested by the presence of repABC regions. This paper contributes to the scarce knowledge on the composition of rhizosphere populations.

Some points that in my opinion would contribute to the paper are:
1 - Authors comment that the soil used has no history of legume cultivation, but, have other plants been cultivated in that soil?
2 - The identification of repABC regions suggests the presence of plasmids. These could be easily visualized by using the approppriate electrophoretic techniques.
3 - If megaplasmids are present, the genomic analysis could be biased by not separating chromosomal from plasmid sequences.

Minor points:
Line 44: change to: ..they have rarely been isolated..

Line 81 - 82: "isolation of non-symbiotic Rhizobia from plant material (Segovia et al., 1991; Sullivan et al., 1996; Van Insberghe et al., 2015; Jones et al., 2016)."
In the references cited, non-symbiotic rhizobia were isolated from rhizospheric soil. Only in the work by Sullivan bacteria were also isolated from the surface of roots. I think this should be clarified.

Line 102: See general comment Nª1

Line 116: "..100 µl of each dilution was plated.." change to: 100 µl of each dilution were plated

Line 131: "primners" change to primers

Lines 135 - 136: "10mM of each dNTP" Shouldn't this be 10 micromolar?

Lines 192 - 195: "while the remaining 53, surprisingly, clustered with sequences from genus Neorhizobium (Fig. 1B). These sequences were compared and classified into groups (twenty in total) if they differed in at least one nucleotide. In order to facilitate further studies, a representative strain of each these rpoB sequence types was chosen"
If I understand correctly, the 20 branches shown in Fig 1B represent the different groups. Table 1 lists 55 strains isolated in this study, but it is not clear which strains belong to each group. Please clarify.
Also, it would be interesting to see which are the variations in the rpoB sequence, maybe in a Supplementary figure?

Lines 206 - 207: "with legume plants were hampered by the fact that the known hosts of Neorhizobium spp. are very diverse and are not found in the Tomejil area"
Which are plants that can be nodulated by Neorhizobium?

Lines 223 - 226: "This gene, together with an upstream ORF, was present in genome assemblies of the Tomejil strains and contained sequences partially complementary to those of fnrN primers that may explain the successful amplifications observed (Fig. S2B).
Was it absent in the Neorhizobium spp. genomic sequences available in databanks, N. galegae and N huautlense?

Lines 231 - 232: "All five genomes contained a set of repABC sequences characteristic of rhizobial plasmids"
See general comment Nª 2

Line 279: "to enrich in soybean-specific" change to: to enrich soybean-specific

Line 291: "we had been able" change to: we were able

Reviewer 2 ·

Basic reporting

In this manuscript Soenens and Imperial describe the isolation and genomic sequences of Rhizobiaceae belonging to a novel non-symbiotic Neorhizobium genospecies. Overall, the reported original research work is sound and the conclusions are supported by the presented data. Below, some points are raised that should be considered in a revised version.

Experimental design

1) The method used to isolate the genetically characterized Neorhizobium strain(s), semi-selective enrichment needs to be clarified for the reader who is not familiar with growth requirements of Rhizobiaceae. I therefore suggest to include the complete recipe of the semi-selective MNBP medium together with the relevant references.
2) For a better understanding of the isolation method the authors may consider to provide Fig.S1 in the main part of the manuscript, not in the supplement. This could also be complemented by indicating numbers (28 isolations, 342 Rhizobium, 99 fnrN positive, 46 Rhizobium, 53 Neorhizobium etc.) to give an overview of how, in the end, 5 draft genome sequences were obtained.

Validity of the findings

1) There is an issue with the PCR amplification using the fnrN primers. A product of the correct size would indicate the presence of the fnrN gene which is indicative of Rhizobium spp. or Agrobacterium spp. However, as shown in Fig.S2, the fragment sizes are clearly different (between the positive control and the isolates). Therefore, the authors erroneously assumed to have isolated fnrN+ strains (53 as the rpoB sequences revealed). That should be clearly stated in the results section (lines 186-193; lines 297-298) with an indication of whether the remaining 46 isolates resulted in a fnrN band of the expected (correct) size.
2) To better understand the false positive fnrN reaction of the Tomejil strains it should be noted how similar the sequences of the primers with the identified target sequence in the sequenced genomes are. Lines 224 to 226 states a „partial complementarity“ which does not explain the positive results in the PCR reaction.
3) The authors report the presence of highly similar repABC sequences in all sequenced Tomejil strains. However, no information is given on other (cargo) genes that are located on the same contigs as the repABC genes. Such an information would provide additional valuable information on the transferability of those plasmids or chromids. At least a scan for DNA transfer genes should be done and the results indicated. Also the possibility that those large elements could represent chromids (see doi:  10.1093/gbe/evv227 and references cited therein) should be considered or discussed.

Additional comments

Minor points:
line 58: …study, Sullivan and colleagues (Ref) were able…..
line 131: …primers…
line 151: …were assembled ….
lines 327-328: please specify what is meant by: …a large diversity of Rhizobial diversity…

---

## Round 0.2 · accepted · Accept

Dear Dr. Imperial,

I am glad to inform you that your article has been accepted for publication in PeerJ.

Best regards,

Elisabeth Grohmann
Academic Editor

# ·

Basic reporting

No comment

Experimental design

No comment

Validity of the findings

No comment

Additional comments

In line 282, the sentence should say: "... replication of a large...."

Reviewer 2 ·

Basic reporting

No further comment

Experimental design

No further comment

Validity of the findings

No further comment

Additional comments

No further comment